# A Conductive Microcavity Created by Assembly of Carbon Nanotube Buckypapers for Developing Electrochemically Wired Enzyme Cascades

**DOI:** 10.3390/nano14060545

**Published:** 2024-03-20

**Authors:** Itthipon Jeerapan, Yannig Nedellec, Serge Cosnier

**Affiliations:** 1Division of Physical Science and Center of Excellence for Innovation in Chemistry, Faculty of Science, Prince of Songkla University, Hat Yai 90110, Thailand; 2Center of Excellence for Trace Analysis and Biosensor, Prince of Songkla University, Hat Yai 90110, Thailand; 3UFR de Chimie et de Biologie, Département de Chimie Moléculaire, Université Grenoble Alpes, CNRS, DCM UMR 5250, F-38000 Grenoble, France; yannig.nedellec@univ-grenoble-alpes.fr; 4Centre for Organic and Nanohybrid Electronics, Silesian University of Technology, Konarskiego 22B, 44-100 Gliwice, Poland; 5Department of Physical Chemistry and Technology of Polymers, Silesian University of Technology, M. Strzody 9, 44-100 Gliwice, Poland

**Keywords:** enzyme cascade, buckypaper, glucose oxidase, horseradish peroxidase, glucose

## Abstract

We describe the creation of a conductive microcavity based on the assembly of two pieces of carbon nanotube buckypaper for the entrapment of two enzymes, horseradish peroxidase (HRP) and glucose oxidase (GOx), as well as a redox mediator: 2,2′-azino-bis(3-ethylbenzothiazoline-6-sulfonic acid diammonium salt (ABTS). The hollow electrode, employing GOx, HRP, and the mediator, as an electrochemical enzyme cascade model, is utilized for glucose sensing at a potential of 50 mV vs. Ag/AgCl. This bienzyme electrode demonstrates the ability to oxidize glucose by GOx and subsequently convert H_2_O_2_ to water via the electrical wiring of HRP by ABTS. Different redox mediators (ABTS, potassium hexacyanoferrate (III), and hydroquinone) are tested for HRP wiring, with ABTS being the best candidate for the electroenzymatic reduction of H_2_O_2_. To demonstrate the possibility to optimize the enzyme cascade configuration, the enzyme ratio is studied with 1 mg HRP combined with variable amounts of GOx (1–4 mg) and 2 mg GOx combined with variable amounts of HRP (0.5–2 mg). The bienzyme electrode shows continuous operational stability for over a week and an excellent storage stability in phosphate buffer, with a decay of catalytic current by only 29% for 1 mM glucose after 100 days.

## 1. Introduction

Bioelectrodes that harness the combined capability of many oxidoreductase enzymes at the same time to catalyze multistep reactions are important for biosensing applications [1,2] and bioconversion of energy [3,4]. Systems that integrate multiple enzymes have been developed for biochemical analysis and for optimizing the profitability of substrate consumption [5,6]. This is achieved by maximizing the number of electrons during the oxidation or reduction of the substrate molecule. One-pot reaction based on enzyme cascades in a single platform enables the deep oxidation of biofuels or the production of value-added products without the need to separate intermediates [7,8]. Recently, artificial multienzyme systems have been used to construct bioelectrodes that include an integrated one-pot catalytic system [1,2,9,10]. In addition to biosensing applications, enzyme cascades enable the full oxidation of fuels, which is essential for effective biofuel cells [3,11]. However, the optimization of this type of electro-enzymatic system is difficult and time-consuming to set up. Indeed, the association of several enzymes with different specific activities in precise ratios is complicated to achieve. The immobilization of enzymes is critical for the advancement of bioelectronics, as it enhances their durability on bioelectronic interfaces, enabling their continuous use [12,13,14]. To immobilize multiple enzymes, it is necessary to ensure compatibility between enzymes, establish efficient electron transfers, maintain catalytic activity, and optimize support materials. However, the immobilization step can affect the optimal conformation of each enzyme and decrease the accessibility of the substrate to the enzymes. The immobilization of multi-enzyme systems on electrodes has been undertaken using a variety of methods including adsorption, covalent bonding, trapping, and cross-linking [1]. The process of enzyme immobilization presents several challenges [15], particularly the low amount of enzymes that can be fixed. The effectiveness of immobilization can also be limited by the availability of reactive sites on the support surface. In addition, tuning specific ratios of enzymes on a surface can be challenging, especially when developing biocatalytic systems that require specific ratios. In particular, the creation of a multienzyme system requires a different grafting process for each enzyme or even trapping of enzyme mixtures. In addition, given the limited surface area of the electrode, the immobilization of several enzymes reacting in a cascade involves reducing the quantity of enzyme catalyzing the first reaction. Therefore, more electrons are generated per molecule, but the initial oxidation related to the amount of the first step enzyme will decrease. Moreover, it is sometimes necessary to add redox mediators to electrically wire one or more enzymes to the electrode. Taking into account the fact that the redox mediators must also be fixed, this reduces the possibilities of enzyme grafting and above all complicates the grafting step, which must be selective for each enzyme. In the case of enzyme trapping in materials such as hydrogels, polymers, etc., the problem lies in the diffusion of the substrate and its successive degradation products in the electrode biomaterial.

An original alternative involves combining the advantages of enzymes in solution, such as accessibility and ease of modulating ratios, with the benefits of immobilization on an electrode, particularly the requirement for a small quantity of enzyme. In nature, enzymes are contained and concentrated within nanocompartments. Enzyme cascades can be highly effective, achieving high reaction rates and minimizing intermediate losses. Despite progress in the development of cascade biocatalytic systems, the realization of a simple and rapid strategy for tuning multiple enzyme ratios within electrochemically wired enzyme cascades has been overlooked. To the best of our knowledge, existing solutions have yet to target such goals as creating conductive microcavities based on carbon nanotube buckypaper for the exploitation of enzyme cascades in microvolumes. Therefore, our goal was to create a model of a carbon nanotube-material-based electrode with a microcavity containing enzymes, where the structure is permeable to water and enzyme substrates but impermeable to enzymes. The design was achieved by assembling two buckypapers, which facilitated the formation of the microcavity and the electrical connection of the multienzymatic system.

To illustrate this new concept of electrode fabrication based on a cascade of enzymes involving redox mediators, a widely studied model of the association of two enzymes, glucose oxidase (GOx) and peroxidase (HRP), was chosen. This model system was combined with redox mediators for the electrical wiring of the HRP (Figure 1). The modulation of the enzyme ratio was achieved simply by depositing a controlled amount of each enzyme in powder form in the microcavity before its closure. Using GOx and HRP as a model system, glucose was oxidized, and the generated H_2_O_2_ was reduced by the wired HRP. In addition to the study of different redox mediators in terms of operational stability and efficiency to connect HRP, the ratio of the two entrapped enzymes was examined to maximize the amplitude of the amperometric signal to glucose.

## 2. Materials and Methods

### 2.1. Chemicals and Reagents

Multi-walled carbon nanotubes (MWCNTs) were obtained from Nanocyl (Sambreville, Belgium). These are multi-walled tubes with a diameter of 9.5 nm, a length of 1.5 μm, and a purity greater than 95%. GOx was obtained from *Aspergillus niger* (163.4 units/mg), peroxidase from horseradish (HRP, 200 units/mg), and 2,2′-Azino-bis(3-ethylbenzothiazoline-6-sulfonic acid diammonium salt (ABTS), hydroquinone, potassium hexacyanoferrate (III), glucose, H_2_O_2_, pyrogallol, and dimethylformamide (DMF) were obtained from Sigma-Aldrich (Saint-Quentin-Fallavier Cedex, France). All chemicals were used without further purification. Sodium phosphate buffer or PB (pH 7.0) was prepared by mixing sodium phosphate monobasic (NaH_2_PO_4_·H_2_O, 42.23 mmol) and sodium phosphate dibasic (Na_2_HPO_4_·7H_2_O, 57.77 mmol), dissolving in water, diluting to 1.0 L, and adjusting the pH to 7.0 using a pH meter (Hach Lange GmbH, Düsseldorf, Germany) and a suitable acid or base solution. The buffer solution was stored at 4 °C for later use. Glucose solutions were kept in the refrigerator for 24 h for mutarotation.

### 2.2. Electrochemical Measurements

Electrochemical experiments were conducted using a conventional three-electrode cell. The working electrode was a sandwich electrode made from buckypaper. A Pt wire served as the counter electrode. The potential was measured relative to an Ag/AgCl (saturated KCl) reference electrode. The experiments were performed at room temperature in a cell containing 0.1 M PB (pH 7.0). Glucose or H_2_O_2_ was added into the electrolyte for investigating the responses toward glucose or H_2_O_2_. Cyclic voltammetry and amperometric measurements were conducted using a potentiostat (Princeton Applied Research PARSTAT-MC-PMC1000, AMETEK, Inc. (Berwyn, PA, USA) (NYSE: AME)) controlled by Versastudio software, ver. 2.60.6. The calibration curves were obtained by increasing the substrate concentration stepwise under a stirring condition.

### 2.3. Preparation of Buckypaper

To prepare buckypaper, a combination of vacuum filtration and ultrasonication was used. First, 66 mg of MWCNTs was dispersed in 66 mL of DMF through vigorous mechanical shaking for 5 min and ultrasonication for 90 min in an ultrasonic water bath (Bandelin Sonorex RK100, Berlin, Germany). The dispersed MWCNTs in DMF were subsequently vigorously shaken and transferred onto a membrane filter (Millipore polytetrafluoroethylene (PTFE) filter, JHWP, 0.45 μm pore size, Merck Chimie SAS, Fontenay sous Bois, France) by vacuum filtration (MZ 2C NT model, Vaccubrand, Wertheim, Germany). The resulting film was then washed with DMF and water, respectively. The buckypaper was then left to be dried with the assistance of a vacuum pump for 2 h. Finally, the buckypaper was then dried overnight at room temperature.

### 2.4. Preparation of a Buckypaper-Based Sandwich Containing HRP

The MWCNT buckypaper was cut into 13 mm diameter disks and coated with carbon paste around the circumference of the cut buckypaper to create a thin cylinder, confining redox molecules and HRP. A flattened stainless-steel wire (150 μm thick) was attached to the conductive carbon paste (LOCTITE EDAG 423SS E&C; Henkel France S.A.S, Boulogne-Billancourt, France), and the electrode’s edge was sealed with a silicon insulator. HRP powder and a redox mediator were deposited into the cavity before the two buckypaper pieces were sealed. Three different redox molecules, namely ABTS, hexacyanoferrate (III), and hydroquinone, were used to prepare different electrodes. Each electrode was loaded with 1 mg of HRP and 0.5 mg of the redox mediator inside the microcavity. A cross-section image captured by laser-assisted optical microscopy, illustrating the microcavity between the two buckypapers of the electrode, is presented in Appendix A. Additionally, a control electrode containing HRP but no redox mediator was prepared similarly. The electrodes were air-dried for 6 h and then soaked in 0.1 M PB, pH 7.0 before use.

### 2.5. Preparation of a Buckypaper-Based Sandwich Containing Two Enzymes

A buckypaper-based sandwich containing two enzymes was prepared using the same principle as explained in the previous subsection regarding the preparation of a buckypaper-based sandwich containing HRP (Section 2.4). The MWCNT buckypaper was cut into 13 mm diameter disks (Figure 2A) and coated with carbon paste on the circumference of the buckypaper disk to form a thin cylinder for confining redox molecules and biocatalysts (HRP together with GOx) (Figure 2B). The optimal amount of carbon paste used was about 55 ± 12 mg to allow for good reproducibility in the sealing. A flattened stainless-steel wire (150 µm thick) was fixed to the conductive carbon paste, and the edge of the electrode was sealed with the silicon insulator. Enzyme powder (HRP and GOx) and a redox mediator were deposited in the cavity before it was closed, and the electrodes were air-dried for 6 h. The electrode was then soaked in 0.1 M PB, pH 7.0 before use.

### 2.6. Determination of the Presence in Solution of Enzymes Due to the Loss of Enzymes from the Microcavity

HRP activity was determined using a UV-Visible spectrophotometric method with pyrogallol and H_2_O_2_ as substrates, following a modified procedure [16]. The bioelectrode containing HRP was immersed in 10 mL of stirred 0.1 M PB, pH 7.0. After 1 day and 5 days, 0.5 mL of the resulting solution was mixed with 0.5 mL of PB containing 10 mM H_2_O_2_ and 8 mM pyrogallol. Enzymatic activity was assessed by monitoring the increase in absorbance at 420 nm due to purpurogallin production from pyrogallol in the presence of H_2_O_2_ and released HRP. The presence of GOx was estimated by directly measuring the absorbance at 278 nm, characteristic of the FAD cofactor, in the incubation solution.

## 3. Results and Discussion

### 3.1. Study of Redox Mediators for Conjugation with HRP

The combination of GOx and HRP has been widely used either to develop glucose biosensors or to elaborate biofuel cells [17,18]. In this work, the enzyme cascade model involved the sequential reactions of two enzymes as well as a mediated electrical connection step of one enzyme. This model configuration was chosen to also show that the hollow electrode could contain and retain, in addition to enzymes, electroactive organic or organo-metallic compounds. First, GOx converts β-D-glucose to gluconolactone and produces H_2_O_2_ which is converted to water by HRP. The latter is electrically wired to the electrode by a redox mediator (Figure 1).

HRP has been extensively studied due to its stability and availability for the development of biosensors and diagnostic tests [19]. In particular, HRP has been widely applied to the development of bienzyme electrodes through its combination with other oxidase enzymes [1,20,21]. Thus, the catalytic activity of HRP can be converted into an electrochemical signal by transferring electrons from the electrode to the oxidized form of HRP. However, the direct transfer of electrons between the electrodes and the active heme group of HRP is generally a slow process. To overcome this drawback, redox mediators are frequently used to construct peroxidase electrodes [22,23]. Although electrodes based on carbon nanotubes showed the possibility of establishing a direct or mediated electrical connection with the enzyme, these configurations were based on immobilized HRPs or even on fixed redox mediators [24]. To obtain an efficient electrical connection of peroxidases, an optimal orientation of the immobilized enzyme is necessary as well as the accessibility of the immobilized redox mediator to the active site of HRP [25]. To overcome these constraints, an alternative is to use the mediator and HRP in solution. In this work, we, therefore, studied different systems based on mediators and the enzyme in solution in the microcavity to determine the most appropriate mediator to improve electron transfer.

It is important to identify a suitable redox mediator to facilitate electron communication within the nanomaterial-based microcavity. The redox mediators used in this study include ABTS, hydroquinone, and hexacyanoferrate (III). These mediators are commonly employed for effectively connecting peroxidases. In the microcavity of the hollow electrodes, these different redox mediators were co-entrapped with HRP to investigate the wiring effect of three redox mediators. The resulting bioelectrodes exhibited the conventional electrochemical behavior expected for the entrapped redox mediators with electrochemically quasi-reversible systems at *E*_1/2_ = 474 mV, *E*_1/2_ = 57 mV, and *E*_1/2_ = 65 mV vs. Ag/AgCl for ABTS, hydroquinone, and hexacyanoferrate (III), respectively. Such mid-potential values of the redox couple peaks observed on the buckypaper sandwich electrodes align with those reported in the literature (ABTS [26], hydroquinone [27,28,29], and hexacyanoferrate (III) [30]). This confirms the penetration of water into the microcavity and the solubilization of redox compounds. The efficiency of the HRP wiring was investigated by recording the amperometric current responses of these bioelectrodes at −0.20 V under hydrodynamic conditions for successive injections of H_2_O_2_ in 0.1 M PB (pH 7.0). Figure 3 shows the linear part of the calibration curves reflecting the current-concentration response of the bioelectrodes as a function of H_2_O_2_ concentration. The control electrode containing only HRP and without any mediator showed quasi-insensitivity towards H_2_O_2_. These results demonstrated that the addition of freely moving mediators played a crucial role in facilitating the bioelectrochemical reduction of H_2_O_2_. It appears that ABTS was the most effective mediator for connecting HRP, with the bioelectrode displaying a sensitivity of 924 ± 22 μA mM^−1^. This sensitivity is much higher than those recorded for hydroquinone and hexacyanoferrate (III), namely 89 ± 4 and 108 ± 12 μA mM^−1^, respectively. The HRP-based hollow electrode without the redox mediator showed the lowest cathodic responses to increasing H_2_O_2_ concentrations, confirming the necessity of employing redox mediation to facilitate the electrochemical reduction of H_2_O_2_ with the bioelectrocatalytic HRP system.

With the aim of developing multi-enzymatic systems involving the electrical connection of one or more enzymes by redox mediators, the efficiency of the physical trapping of these mediators in the microcavity was examined by cyclic voltammetry over several months. Figure 4 shows the cyclic voltammograms for the ABTS-containing electrode and the charges recorded for the three redox bioelectrodes after 3, 5, and 7 months of storage in PB. It appears that the intensity of the redox couples increases initially then decreases slightly (4 to 25%) between 3 and 7 months. Despite the porosity of the buckypapers, these results regarding the maintenance of high current for the redox couples confirm the effective trapping and retention of electroactive compounds within the microcavity generated within high-surface-area nanomaterials over the long term. In addition, the electrochemical behavior of the different redox electrodes containing HRP has been characterized in the presence and absence of H_2_O_2_. Surprisingly, in the presence of H_2_O_2_, a catalytic current due to the electroenzymatic reduction of H_2_O_2_ is still present at 0 V for the three redox electrodes containing HRP after 7 months. Three such HRP-based electrodes contain different redox molecules: ABTS, hexacyanoferrate (III), and hydroquinone. The absolute values of the cathodic current intensity, obtained by subtracting the current in the absence of glucose during cyclic voltammetry scans after 3, 5, and 7 months with hydroquinone, were 231 ± 15 μA, 318 ± 22 μA, and 337 ± 10 μA, respectively. For hexacyanoferrate (III), under identical conditions, the absolute current values of the bioelectrocatalytic response after 3, 5, and 7 months were 181 ± 7 μA, 254 ± 8 μA, and 184 ± 6 μA, respectively. The highest catalytic current after 7 months (≈528 ± 10 μA) was recorded with ABTS, corroborating the better electrical connection of HRP by ABTS. This also shows that enzymatic activity can be observed after 7 months of storage in a buffer for this confined enzyme. It should be noted that a slight positive shift in the onset potential of H_2_O_2_ reduction appeared with storage time in PB. Taking into account that HRP and ABTS are in powder form, this phenomenon may be ascribed to the slow penetration of water inside the cavity. As a result, initially, the two components are not completely dissolved which could increase the steric hindrances and/or the resistance in the electrolytic solution. The ABTS redox system would, therefore, be less reversible and less intense initially than after several months of immersion in PB.

### 3.2. Determination of Optimal Ratio for Two Enzymes

Two sheets of freestanding buckypapers were assembled to form a conductive platform immobilizing HRP, GOx, and ABTS (as a redox mediator). The optimal ratio of the two enzymes is one of the most important parameters in the performance of the bioelectrode. First, GOx catalyzes glucose oxidation to produce inside the buckypaper microcavity gluconolactone and H_2_O_2_ which is further reduced by HRP. The oxidized form of HRP is then reduced by ABTS, thus generating a cathodic current which can be proportional to the glucose concentration in the buffer solution. Therefore, the effect of varying the ratio of HRP/GOx on the performance of the biosensor was investigated by recording the reduction current of ${ABTS\cdot }^{+}$ at 50 mV vs. Ag/AgCl as a function of glucose concentration. To assess the impact of different amounts of GOx on the electrochemical performance of the bioelectrodes while maintaining a fixed amount of HRP (1 mg), amperometric analysis was performed with glucose concentrations ranging from 30 μM to 20 mM (Figure 5). It appears that an increase in the GOx amount from 1 to 2 mg resulted in higher sensitivity and maximum current (*I_max_*) due to the faster rate of H_2_O_2_ production, which would imply that the bioelectrode response is limited by GOx activity. However, increasing the amount of GOx from 2 to 3 mg and 4 mg induces a slight decrease in sensitivity associated with a marked decrease in maximum current. The increase in the quantity of GOx leads to a greater production of H_2_O_2_ which inhibits the activity of HRP and consequently decreases the maximum current [31].

Therefore, for a fixed quantity of 1 mg HRP, equivalent to 200 HRP units (or 200 µmole ABTS/min), the optimal amount of GOx to achieve both maximum sensitivity and maximum current is 2 mg, corresponding to 327 µmole glucose/min, theoretically necessitating 654 units of HRP. Consequently, this indicates that the optimal GOx-to-HRP weight and unit ratios are 2:1 and 3.6:1, respectively. This phenomenon could suggest a rapid loss of H_2_O_2_ by diffusion towards the outside of the microcavity.

To evaluate the retention of both enzymes within the microcavity of the optimized bioelectrode configuration, we investigated the potential release of enzymes from the trapped microcavity, where the bioelectrode was incubated in a 0.1 M PB (pH 7.0) for durations of 1 day and 5 days. The release of HRP was monitored by assessing its enzymatic activity via absorbance measurements at 420 nm, which indicates purpurogallin production from pyrogallol in the presence of H_2_O_2_. The absence of spectrometric evolution confirms the effective retention of the trapped enzyme molecules in the microcavity. Given the smaller molecular size of HRP (44 kDa [32]) compared to GOx (160 kDa [33]), it is difficult for the latter to diffuse through the buckypapers. Additionally, direct spectrometric analysis of the incubation solution after 5 days revealed no absorbance at 278 nm, a characteristic of the FAD cofactor of GOx [34], thus confirming the retention of this enzyme.

Moreover, the effect of varying amounts of HRP on the performance of bioelectrodes containing a fixed amount of 2 mg of GOx was also investigated (Figure 6). For 0.5 mg HRP, the response of the bioelectrode is limited by the activity of HRP and its inhibition by H_2_O_2_. Increasing the amount of HRP from 0.5 mg to 1 and then 1.5 mg leads to a marked improvement in the sensitivity and *I_max_* of the bienzymatic system. On the other hand, the increase in HRP to 2 mg induces a decrease in sensitivity and *I_max_*, corroborating the optimum value of the GOx/HRP ratio by weight between 2:1 and 1.33:1. To validate the cascade operation of the two enzymes, a control experiment was carried out with a hollow electrode containing 2 mg of GOx and 1 mg of ABTS and without HRP. As expected, the resulting bioelectrode provided no response to glucose at 50 mV vs. Ag/AgCl, supporting the need to have the association of HRP and GOx for glucose sensing.

The reproducibility of the bioelectrode elaboration based on 1 mg HRP and 2 mg GOx was evaluated from the maximum current values in the presence of glucose with three different electrodes. The resulting relative standard deviation (RSD) to *I_max_* determination was 8%.

### 3.3. Stability Assessment of GOx, HRP, and ABTS-Based Bioelectrodes

To assess the operational stability of the bioelectrode based on GOx, HRP, and ABTS, a potentiostatic test was carried out at 50 mV vs. Ag/AgCl in glucose solution. The electrochemical cell was connected to peristaltic pumps to provide a constant flow of 20 mM glucose in 0.1 M PB (pH 7.0) at a rate of 4 mL/h (Figure 7A). After an initial decrease in the signal of approximately 6 μA/h for 48 h, a quasi-stabilization of the cathodic current is recorded between 48 and 196 h, the decrease being only 0.5 μA/h. At 192 h, the reduction current value dropped to 63 μA, or 16% of its initial value. To confirm the origin of the residual cathodic current after 165 h, the glucose solution was replaced by a buffer without glucose. The removal of glucose led to a drastic drop in the catalytic current from 71 to 15 μA. After 2 h, the reintroduction of 20 mM of glucose induced the restoration of the catalytic current (69 μA), illustrating the continuous operation of the bienzymatic cascade (Figure 7B). This observation also confirms that the bioelectrocatalytic cathodic current arises from the enzymes trapped within the microcavity and not from enzymes released into the analysis solution.

The storage stability of the GOx/HRP bienzymatic electrode was also evaluated in 0.1 M PB, pH 7.0, at 4 °C, over a period of 100 days by periodically measuring its current response to 1 and 20 mM glucose. It appears that the current response for 1 and 20 mM glucose retains 71% and 39% of its initial value, respectively, after 100 days of storage in 0.1 M PB (Figure 8). The difference in stability observed for glucose concentrations of 1 and 20 mM may be ascribed to an inhibition process or a denaturation of the HRP protein by the production of a higher H_2_O_2_ concentration at 20 mM glucose.

To investigate the nanomaterial’s morphology, the surface of the buckypaper was characterized immediately after the filtration step in the buckypaper synthesis process. It was observed that the side in contact with the PTFE filter appeared smoother and more regular compared to the side in contact with the solution to be filtered (Appendix A). The rougher side of the two buckypapers was used to form the interior of the cavity, while the smoother side was positioned on the exterior. The evolution of the buckypaper surface inside the cavity, which contained enzymes (2 mg GOx and 1 mg HRP), was also investigated after 9 months of immersion in 0.1 M PB (pH 7.0). Upon opening the cavity, the interior surface of the buckypaper appeared almost identical to the initial surface. However, in terms of the external surface, prolonged contact with water slightly increased surface roughness (Appendix A(A’,B’)). This evidence supports the robustness of the lab-made carbon-nanotube buckypaper in retaining the microstructural features critical for the long-term stability of the cascade system within the microcavity.

## 4. Conclusions

We have described an original concept for the fast elaboration of enzyme cascade reactions in the confined space of hollow planar electrodes based on the assembly of two buckypapers. The complexity of optimizing a multienzyme system illustrates the interest in being able to easily modulate the ratio of enzymes simply by changing the masses of trapped enzymes. In addition to the easy modulation of enzyme ratios, we have also demonstrated the possibility of trapping with enzymes a redox mediator, ensuring the electrical connection of an enzyme. Following this strategy, the immobilization of HRP and GOx in a buckypaper microcavity, along with a redox mediator (i.e., ABTS), led to an efficient bioelectrode for glucose sensing at 50 mV vs. Ag/AgCl. The bienzyme electrode demonstrated continuous operation for over a week and excellent storage stability, retaining 71% of its initial sensitivity even after 100 days. It is expected that such construction and fast optimization of multienzyme electrochemical systems based on hollow electrodes will be useful for the development of biosensors and biofuel cells.

## Figures and Tables

**Figure 1 nanomaterials-14-00545-f001:**
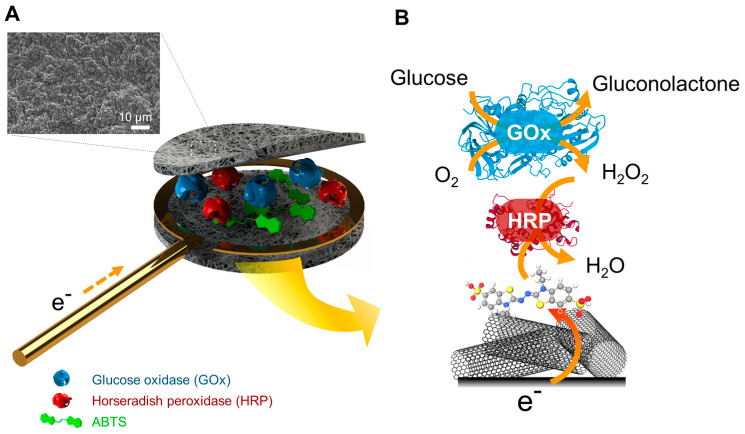
Schematic representation of the bienzyme electrode’s functionality, based on two enzymes and a redox mediator within a buckypaper microcavity, for glucose oxidation and hydrogen peroxide reduction. (**A**) The illustration depicts the exploded three-dimensional structure showing the concept of the microcavity structure. The inset displays a microscopic image of the inner electrode’s side in contact with the enzymes and mediator. (**B**) The enzymatic reactions enabling the recording of cathodic current from the cascade electrode.

**Figure 2 nanomaterials-14-00545-f002:**
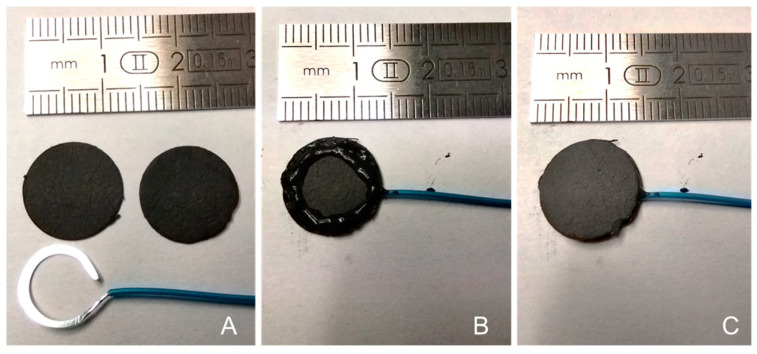
Photographs illustrating electrode cavity fabrication processes: (**A**) Two buckypapers and the flattened contact wire. (**B**) The cavity created by the adhered contact wire with the conductive glue. (**C**) The sealed electrode containing the enzyme and mediator powder.

**Figure 3 nanomaterials-14-00545-f003:**
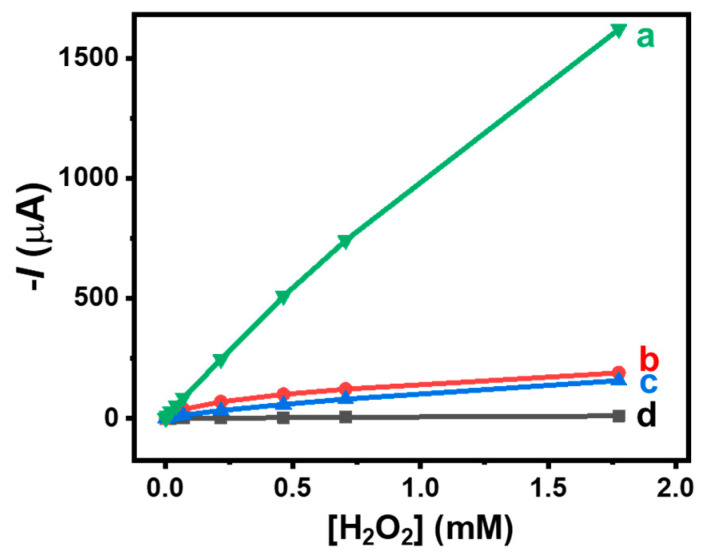
Amperometric current response of HRP (1 mg)-based hollow electrodes as a function of H_2_O_2_ concentration. Bioelectrodes containing (a) ABTS, (b) hexacyanoferrate (III), and (c) hydroquinone, and (d) without redox mediator. Applied potential −0.20 V vs. Ag/AgCl in 0.1 M PB (pH 7.0).

**Figure 4 nanomaterials-14-00545-f004:**
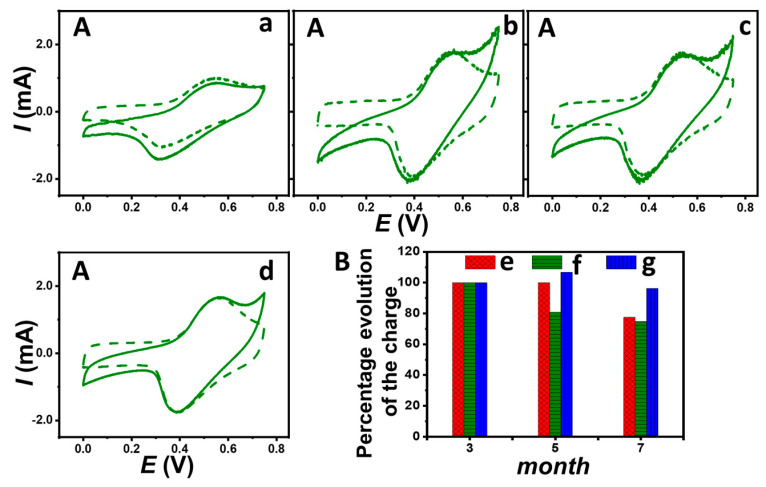
(**A**) Cyclic voltammograms (5 mV s^−1^) of a HRP-based bioelectrode with ABTS in the presence (solid line) and absence (dashed line) of 4 mM H_2_O_2_ recorded in 0.1 M PB at pH 7.0: bioelectrode stored in PB for (**a**) 1 day, (**b**) 3, (**c**) 5, and (**d**) 7 months. (**B**) Percentage evolution of the charge (pic integration) over a 3-, 5- and 7-month storage for HRP-based bioelectrodes containing: (**e**) hexacyanoferrate (III), (**f**) hydroquinone, and (**g**) ABTS.

**Figure 5 nanomaterials-14-00545-f005:**
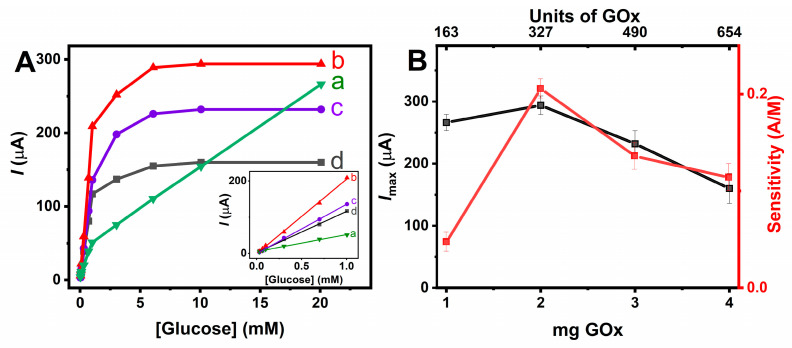
(**A**) Absolute value of the amperometric current response of hollow electrode containing 1 mg HRP and (a) 1, (b) 2, (c) 3, and (d) 4 mg of GOx to glucose concentration; inset linear part of the calibration curves for glucose at bioelectrodes containing different amounts of GOx (1–4 mg). (**B**) Effect of the GOx loading on the maximum current response and glucose sensitivity of the bienzyme electrodes. Error bars represent data variability for three repetitive measurements of the calibration curve for the same bioelectrode. Applied potential 50 mV vs. Ag/AgCl; air-saturated in 0.1 M PB (pH 7.0).

**Figure 6 nanomaterials-14-00545-f006:**
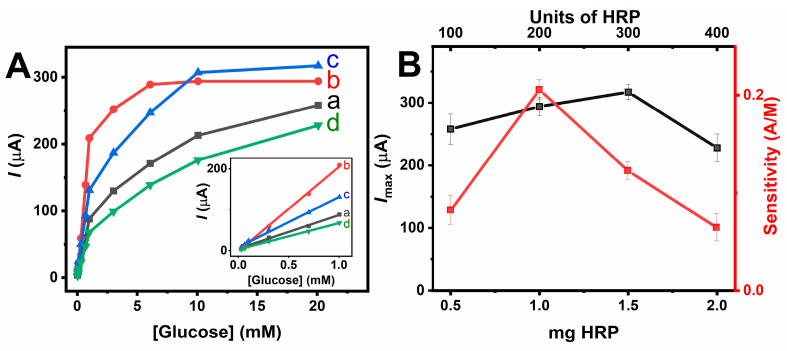
(**A**) Absolute value of the amperometric current response of hollow electrode containing 2 mg GOx and (a) 0.5, (b) 1, (c) 1.5, and (d) 2 mg of HRP to glucose; inset linear response region of the calibration curves for glucose at bioelectrodes containing different amounts of HRP (0.5–2 mg). (**B**) Influence of the HRP amount on the maximum current response and glucose sensitivity of the bienzyme electrodes. Applied potential 50 mV vs. Ag/AgCl; air-saturated in 0.1 M PB (pH 7). Experimental conditions as in Figure 5.

**Figure 7 nanomaterials-14-00545-f007:**
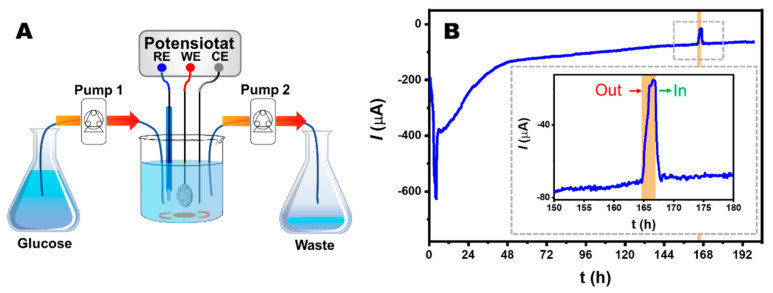
(**A**) Schematic representation of the electrochemical cell connected to two peristaltic pumps providing a constant flow of 20 mM glucose in 0.1 M PB, pH 7.0 at a rate of 4 mL/h. (**B**) Amperometric current response of hollow electrode containing 1 mg ABTS, 1 mg HRP, and 2 mg of GOx to 20 mM glucose; inset bioelectrode response in presence and absence of glucose. Experimental conditions as in Figure 5.

**Figure 8 nanomaterials-14-00545-f008:**
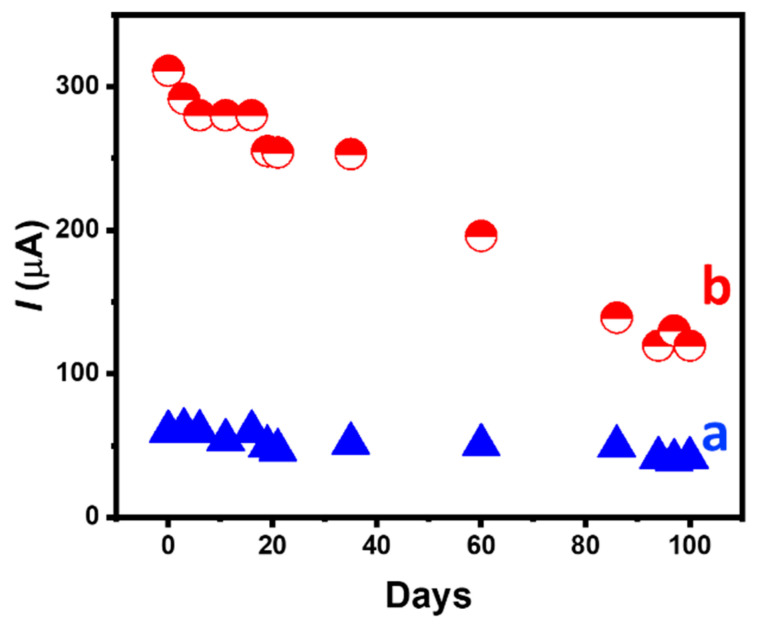
Absolute value of the amperometric current response of hollow electrode containing 1 mg ABTS, 1 mg HRP and 2 mg of GOx to (a) 1 and (b) 20 mM glucose as a function of time. Bioelectrode stored in 0.1 M PB, pH. 7.0. Experimental conditions as in Figure 5.

## Data Availability

The data that support the findings of this study are available with the corresponding author, Serge Cosnier, upon reasonable request.

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
