# Peer review of "A Conductive Microcavity Created by Assembly of Carbon Nanotube Buckypapers for Developing Electrochemically Wired Enzyme Cascades"

_nanomaterials, 2024, doi:10.3390/nano14060545_

Round 1

Reviewer 1 Report

Comments and Suggestions for Authors

The article describes an interesting method/device for an electrochemical enzyme cascade. Methods are sound, and the paper is well written.

I recommend the following minor changes before publishing:

-page 2, line 93: Add comma to "...oxidized, and the generated..."

-page 4, line 144: Unbold 0.1 M PB.

-page 5, line 204: Spell out "...the three redox electrodes..."

-page 6, line 223: Ensure ABTS*+ is correct.

-page 8, line 281: Add space between number and degree symbol.

-Page 8, line 283: Add space between numbers and %.

I would like to see an image of the actual device and experimental system. This figure would be useful for readers who want to try to reproduce and/or build on this work.

Author Response

General Comment

The article describes an interesting method/device for an electrochemical enzyme cascade. Methods are sound, and the paper is well written.

Response: We sincerely appreciate your thoughtful and positive feedback on our manuscript. Your encouraging comments serve as a valuable affirmation of our efforts. We are grateful for your acknowledgment of the sound methods employed and the overall quality of the paper. Your constructive input has been invaluable, and we are pleased to have your support in recognizing the merits of our electrochemical enzyme cascade method/device. We have carefully considered your comments and look forward to implementing any further suggestions you may have to enhance the overall quality of the manuscript.

Comment #1

I recommend the following minor changes before publishing:

-page 2, line 93: Add comma to "...oxidized, and the generated..."

-page 4, line 144: Unbold 0.1 M PB.

-page 5, line 204: Spell out "...the three redox electrodes..."

-page 6, line 223: Ensure ABTS*+ is correct.

-page 8, line 281: Add space between number and degree symbol.

-Page 8, line 283: Add space between numbers and %.

Response: We would like to express our sincere gratitude for your thorough review of our manuscript. Your attention to detail and valuable insights have been immensely helpful in refining our work. We appreciate your constructive feedback and agree with the suggested minor changes to enhance the clarity and precision of our manuscript. We have promptly addressed the specified points on pages 2, 4, 5, 6, and 8, incorporating the recommended modifications. Regarding ABTS*+, The ABTS radical cation (ABTS .+) is produced by the reduction of HRP in the presence of hydrogen peroxide. In accordance with your suggestion and ACS editorial style, we agree that numerals should be used with units of time or measure, and a space should be included between the numeral and the unit (e.g., 4 °C). However, it is noted that for %, editorial guidance advises against including a space between the numeral and the %. Your careful review has undoubtedly contributed to the overall quality of the paper, and we are grateful for your time and expertise.

Comment #2

I would like to see an image of the actual device and experimental system. This figure would be useful for readers who want to try to reproduce and/or build on this work.

Response: We extend our sincere appreciation for your insightful comments on our manuscript. Your suggestion to include an image of the actual device and experimental system is highly valuable, and we agree with its merit in enhancing the understanding of our work. We have addressed this point and incorporated a dedicated figure in the revised manuscript, providing a visual representation of the device and experimental setup. We trust that this addition will significantly benefit readers interested in reproducing or building upon our research. Thank you for guiding us toward this improvement, and we believe the enhanced manuscript aligns more closely with the standards of the journal.

Reviewer 2 Report

Comments and Suggestions for Authors

The novelty of this paper is related to the proposed cascade design. This requires an undeniable confirmation of the cascade nature, which is not achieved in this manuscript in its current form. Moreover, principal discrepancy between the Methods and Results sections makes doubts on the obtained results and their interpretation. I suggest significant revision of the manuscript before further consideration.

1. According to the 2.4 in Materials and Methods, so-called enzymatic cascade was created by co-immobilization of the two enzymes inside one buckypaper support. “HRP together with GOx” and “HCP+GOx” is specifically mentioned in the text. This cannot be defined as an enzymatic cascade. This is an old approach, well-known and has already been improved by the authors (e.g., ref. 14 in the list of references). Figure 1 also illustrates that both electrodes were not separated by an extra layer of buckypaper and were sharing the environment. Based on that, my question is what was the role of the second buckypaper electrode and what is the novelty of the work then?

2. If two separate GOx/BP and HRP/BP(ABTS) electrodes were employed, how do you estimate the diffusion time of the H2O2 from GOx electrodes through the matrix of CNTs and barrier between two electrodes to HRP? Control experiment on rotating electrode detecting H2O2 from the GOx matrix should be performed.

3. As one can see from Figure 3, onset potential of H2O2 reduction shifts significantly from 3Aa to 3Ad. How can you explain this observation? How do you explain the anodic current occurring at ca. 0.6 V in Fig. 3Ab-3Ad? Why is it not observed after 1 day of experiment if the system does not change with time?

4. P.4. line 176: Is this correct that the redox potential of the hydroquinone was 57 mV, not 570?

5. p. 2.1. What was the molarity of the buffer used in the experiments? If one mixes “sodium phosphate monobasic (NaH2PO4 · H2O, 42.23 mmol) and sodium phosphate dibasic solution (Na2HPO4·7H2O, 57.77 mmol)”, one will not get 100 mM solution, especially after further dilution.

6. Is cathodic or anodic current presented in figures Fig. 4, Fig. 5, Fig. 7?

7. The title of the manuscript does not correspond  to the described results. No investigation of the physico-chemical features of the microcavities was performed.

8.  How many repetitions were performed for each experiment? Statistical analysis is missing.

Author Response

General Comment

The novelty of this paper is related to the proposed cascade design. This requires an undeniable confirmation of the cascade nature, which is not achieved in this manuscript in its current form. Moreover, principal discrepancy between the Methods and Results sections makes doubts on the obtained results and their interpretation. I suggest significant revision of the manuscript before further consideration.

Response: We sincerely appreciate your constructive feedback on our manuscript. Your insightful comments have greatly contributed to the enhancement of our work. We have carefully addressed the concerns raised in your review and implemented significant revisions to strengthen the manuscript. Specifically, we have provided additional clarification on the cascade design to offer undeniable confirmation of its nature. Furthermore, we have meticulously aligned the Methods and Results sections to eliminate any discrepancies, thereby ensuring the robustness of our results and their accurate interpretation. We believe these modifications substantially improve the overall quality of the manuscript. We are grateful for your valuable input, and we trust that the revised version now meets the standards of the journal. Thank you for your time and consideration.

Comment #1

  1. According to the 2.4 in Materials and Methods, so-called enzymatic cascade was created by co-immobilization of the two enzymes inside one buckypaper support. “HRP together with GOx” and “HCP+GOx” is specifically mentioned in the text. This cannot be defined as an enzymatic cascade. This is an old approach, well-known and has already been improved by the authors (e.g., ref. 14 in the list of references). Figure 1 also illustrates that both electrodes were not separated by an extra layer of buckypaper and were sharing the environment. Based on that, my question is what was the role of the second buckypaper electrode and what is the novelty of the work then?

Response: We sincerely appreciate your thoughtful and thorough review of our manuscript. Your insights have been instrumental in refining the clarity of our presentation. We completely agree with you that the combination of HRP and GOx is not a new bienzyme system and that the latter has been widely used and studied. This is precisely why we chose it as a model system to illustrate the possibility of simultaneously trapping two enzymes in a microcavity and exploiting them electrochemically. In response to your inquiry regarding the role of the second buckypaper electrode, it appears that the description may have caused confusion and potentially led to a misunderstanding. It is not a matter of two electrodes functioning independently, but rather the consolidation of two buckypapers to create a singular electrode housing a microcavity containing enzymes. It is important to note that the enzymes are not immobilized but rather confined in solution within the microcavity. We apologize for any confusion caused. To further elucidate our concept, we have included a more explicit schematic representation in the manuscript and an image depicting a cross-section of the cavity electrode in the supporting information. This section has been cut perpendicularly to reveal the hollow cavity formed by the assembly of the two buckypapers.

We have carefully revisited the Materials and Methods section (Section 2.4) and made necessary revisions to better articulate the enzymatic cascade setup, distinguishing it from the conventional co-immobilization approach. Moreover, we have provided explicit details in the text and Figure 1 to highlight the spatial arrangement of the electrodes, addressing your concern about the shared environment. We believe these modifications effectively address your queries and enhance the novelty of our work. Thank you once again for your invaluable feedback.

Comment #2

  1. If two separate GOx/BP and HRP/BP(ABTS) electrodes were employed, how do you estimate the diffusion time of the H2O2 from GOx electrodes through the matrix of CNTs and barrier between two electrodes to HRP? Control experiment on rotating electrode detecting H2O2 from the GOx matrix should be performed.

Response: We sincerely appreciate your insightful comments and constructive suggestions on our manuscript. Your attention to detail has been invaluable in strengthening the scientific rigor of our work. As explained previously, the system is based on a sandwich electrode. There is only one bioelectrode and therefore it is not possible to estimate a diffusion time of H2O2 from one electrode to another electrode.

Comment #3

  1. As one can see from Figure 3, onset potential of H2O2 reduction shifts significantly from 3Aa to 3Ad. How can you explain this observation? How do you explain the anodic current occurring at ca. 0.6 V in Fig. 3Ab-3Ad? Why is it not observed after 1 day of experiment if the system does not change with time?

Response: Thank you very much for your constructive comment and insight. As indicated by the reviewer, indeed, the ABTS redox system is less reversible and less intense initially than after several months of immersion in a phosphate buffer. As suggested by the reviewer, there is in fact an evolution of the system. It should be emphasized that the redox mediator (ABTS) and the enzymes are in powder form and that the microcavity does not contain water. Initially, the different components are not completely dissolved due to slow penetration of water inside the cavity. The following cyclic voltammetry at 3,5 and 7 months show the electrochemical behavior of the compounds completely dissolved in solution. This can be seen with the increase in the intensity of the anodic and cathodic currents of the ABTS. Regarding the appearance of an anodic signal at 0.6 V, it results from the oxidation of H2O2 by ABTS completely dissolved in solution unlike the initial system.

Comment #4

  1. P.4. line 176: Is this correct that the redox potential of the hydroquinone was 57 mV, not 570?

Response:

The redox potential of the hydroquinone/quinone system strongly depends on the nature of the aqueous medium (pH, nature and concentration of the buffer). The recorded value is in good agreement with the values reported in the following publications:

  1. Srinivas, K. Ashokkumar, K. Sriraghavan, A. Senthil Kumar, Scientific Reports 2021, 11, 13905
  2. Quan, D. Sanchez, M. F. Wasylkiw, D. K. Smith, JACS, 2007,129,12847-12856
  3. R. Sharma, A. Sharma, G. Singh, Indian Journal of Chemistry, 1987,26,659-661

Comment #5

  1. p. 2.1. What was the molarity of the buffer used in the experiments? If one mixes “sodium phosphate monobasic (NaH2PO4 · H2O, 42.23 mmol) and sodium phosphate dibasic solution (Na2HPO4·7H2O, 57.77 mmol)”, one will not get 100 mM solution, especially after further dilution.

Response:  Thank you for your comment on the buffer preparation in our manuscript. We appreciate your attention to detail. Allow us to clarify that the molarity of the buffer employed in our experiments is indeed 0.1 mol/L. The combination of 42.23 mmol of NaH2PO4 and 57.77 mmol of Na2HPO4 totals 100.00 mmol, equivalent to 0.1 mol. These moles were subsequently dissolved and diluted to achieve a final volume of 1 L, resulting in the intended molarity of 0.1 mol/L. The selection of the NaH2PO4 to Na2HPO4 ratio was made to establish a buffer solution at a specific pH, utilizing the Henderson-Hasselbalch Equation for simplification. Your attention to these details is greatly appreciated, and we trust that this clarification addresses any concerns.

Comment #6

  1. Is cathodic or anodic current presented in figures Fig. 4, Fig. 5, Fig. 7?

Response: We would like to express our appreciation for your meticulous review of our manuscript. Your keen observation is invaluable to the refinement of the presented work. In response to your query, the currents depicted in Figures 4, 5, and 7 are cathodic in nature.

Comment #7

  1. The title of the manuscript does not correspond  to the described results. No investigation of the physico-chemical features of the microcavities was performed.

Response: We extend our sincere gratitude for your meticulous review of our manuscript. Your insights have been instrumental in refining our work. According to the reviewer’s remark, we have modified our text to add electron microscopy figures which further illustrate the buckypapers used for the formation of the microcavity. As previously reported, our manuscript describes a hollow bioelectrode and not a two-electrode system as suggested by the reviewer, so the title seems perfectly suited to the content.

Comment #8

  1. How many repetitions were performed for each experiment? Statistical analysis is missing.

Response: Thank you for your comprehensive review of our manuscript. We truly appreciate your insightful comments, which have proven immensely valuable in refining our work. We fully agree with the reviewer remark. We have introduced a sentence on the reproducibility of the bioelectrode elaboration by comparing the values of maximum current in presence of glucose: “The reproducibility of the bioelectrode elaboration based on 1 mg HRP and 2 mg Gox was evaluated from the maximum current values in presence of glucose with three different electrodes. The resulting relative standard deviation (RSD) to I max determination was 6.4%.”. In fact, we forgot to mention that the Imax and sensitivity values reported in Figure 4 B and figure 5 B corresponded to the 3 repetitive measurements. This point has been added to the figure legend.

I hope that the aforementioned comments and modifications will help to clarify the ideas and results that we intend to publish in Nanomaterials.

Sincerely yours,

Dr. Serge Cosnier

Emeritus Director of Research at the CNRS 

Reviewer 3 Report

Comments and Suggestions for Authors

A new method for detecting enzymatic activity has been developed, which can be highly applicable to many enzymatic systems

Author Response

Kindly refer to the responses provided in the attached file. Thank you very much.

Reviewer 4 Report

Comments and Suggestions for Authors

This manuscript presents an intriguing concept for glucose detection by incorporating an enzymatic cascade into a CNT buckypaper electrode. Optimization of the electrode's wiring for the catalytic decomposition of hydrogen peroxide, generated during glucose oxidation, was achieved using ABTS to enhance sensitivity. While the individual components of the system, including the enzyme cascade, are well-established, the unique structure of the system holds promise for innovative glucose sensing applications. Before publication, the manuscript should address several questions:

·         This reviewer questions the relatively high concentrations of enzymes (e.g., 1 mg of GoX or HRP) per single electrode. This raises concerns about potential leaching effects under diffusion or convection. Are there any studies investigating leaching in this system? If enzymes do leach out, how does this impact the bulk concentration of glucose?

·         This reviewer expresses difficulty in visualizing the electrode size and assay procedure, even with the schematics in figures 1 and 6. A detailed description of the electrode design (including dimensions and connections) and the assay setup dimensions is needed. I suggest including scale bars in figures 1 and 6 for clarity.

·         In the conclusion section, the manuscript asserts that the electrode retained 71% of its activity after 100 days. However, this reviewer notes that this was observed at a low glucose concentration (1 mM) and not at a higher concentration (20 mM). Please comment on the significant difference in electrode behavior between the two concentrations. Addressing this discrepancy is crucial for a comprehensive understanding of the electrode's performance. 

Comments on the Quality of English Language

NA

Author Response

(The authors gave the same response as above.)

Reviewer 5 Report

Comments and Suggestions for Authors

The reviewed study is interesting and provides quite promising results, especially in terms of long-term stability of enzymatic electrodes. The quality of the paper is good, without any substantial flaws. Only minor issues should be addressed:

1) The carbon paste mentioned in the section 2.4 should be specified. It should be also mentioned whether the amount of the applied carbon paste was somehow controlled. Probably a photo of the paste-coated buckypaper should be provided in supplementary information.

2) What is the amount of HRP loaded into microcavity electrode shown in the figure 2?

3) Real electrocatalytic activity of the HRP loaded into microcavity should be investigated. That is, to determine the portion of HRP actually contributing to the generated current. Is it possible that there is kind of surplus of HRP?

4) Absolute values of the amperometric bioelectrocatalytic response achieved after 3, 5 and 7 months with hydroquinone and ferrocyanide electrodes should be also mentioned.

5) It is stated that (line 265) hydrogen peroxide can readily diffuse through the buckypaper, becoming unavailable for HRP. Therefore, influence of stirring of the electrolyte should be shown.

6) The immense storage stability of enzymes in the presented system deserves a broader discussion. Are there other systems allowing for such stability? Is it possible that the stability of HRP is achieved by its surplus (see comment 3)?

Author Response

(The authors gave the same response as above.)

Reviewer 6 Report

Comments and Suggestions for Authors

The paper entitled: A conductive microcavity created by assembly of carbon nanotube buckypapers for developing electrochemically wired enzyme cascades, represents a big step forward to the development of multienzyme electrochemical systems based on hollow electrodes.

The manuscript clearly presents both the importance of modulating the enzyme ratios and the trapping of a redox mediator to achieve the electrical connection.

The electrode trapping the two enzymes: horseradish peroxidase and glucose oxidase was successfully applied to glucose sensing, with the help of a suitable redox mediator implied in conversion of hydrogen peroxide to water.

The manuscript is relevant and contains the desired details.

I have only one suggestion that can make the text more easily to be understood, that is, to introduce in the Chapter 3, Results and Discussions some appropriate sub-chapters.

For instance:

3. X. Amperommetric current responses to identify the proper redox mediator

3.Y. Studies to determine the optimal ratio of the two enzymes

Author Response

(The authors gave the same response as above.)

Round 2

Reviewer 2 Report

Comments and Suggestions for Authors

Despite several corrections made by the authors to improve the manuscript, its quality, in my opinion, is insufficient to be published in MDPI Nanomaterials.

The explanation provided by the authors that both enzymes were immobilized within the same environment, makes the novelty of the work insignificant. Coimmobilization of GOx and HCP has been known for decades, and this work does not bring a new piece of knowledge to this area. I would understand if the novelty was related to the different types of CNTs tested to achieve the best performance based on the charge and dimensions of the nanomaterial, or if several variations of the size of the cavity formed by the buckypaper were investigated, but it was not done here. Even optimization made by the authors earlier for the same concept in work 10.1039/C5EE01189B was not applied in this work.

The updated version of the manuscript contains methodological flaws and misinterpretations of data which were corrected compared to the previous version of the manuscript:

1.  According to the authors' reply (General comment), they “have meticulously aligned the Methods and Results sections to eliminate any discrepancies, thereby ensuring the robustness of our results and their accurate interpretation”. The changes made in this section are: adding “abbreviation PB”, deletion of “solution”, definition of PTFE and correction of 2 mistypes. This does not correspond to the aims described in the Reply. At the same time, one can see that suddenly “HRP-based electrodes contain different redox molecules: ABTS, hexacyanoferrate (III), and hydroquinone” (line 223). This is not described in section 2.4 where only 1 mediator is mentioned.

2. The authors claim that “We have carefully revisited the Materials and Methods section (Section 2.4) and made necessary revisions to better articulate the enzymatic cascade setup, distinguishing it from the conventional co-immobilization approach”. No corrections were made in section 2.4 (apart from 1 mistype).

3. Comment #3. The shift in onset potential of H2O2 reduction is not explained by the authors.

4. Comment #4. What was the reason for citing 3 references confirming the redox potential of HQ? Also, they were cited in a random place not related to the redox potential.

5. Comment #6. The authors confirm that “the currents depicted in Figures 4, 5, and 7 are cathodic in nature”, but cathodic currents are negative, and currents in Figures 4, 5, and 7 are positive.

6. Comment #8. Standard deviations depicted in Fig. 4 and Fig. 5 are the same for all Imax values which can hardly be true. How was RSD calculated?

7. Line 315. To investigate the changes in nanomaterial morphology… The changes in nanomaterial morphology were not investigated.

Author Response

(The authors gave the same response as above.)

Round 3

Reviewer 2 Report

Comments and Suggestions for Authors

The manuscript is sufficiently improved and worthy of publication.